# The Short- and Long-Run Impacts of Air Pollution on Human Health: New Evidence from China

**DOI:** 10.3390/ijerph20032385

**Published:** 2023-01-29

**Authors:** Yayun Ren, Jian Yu, Guanglai Zhang, Chang Zhang, Wenmei Liao

**Affiliations:** 1School of Economics, Guizhou University of Finance and Economics, Guiyang 550025, China; 2School of Economics, Jiangxi University of Finance and Economics, Nanchang 330013, China; 3School of Finances, Zhongnan University of Economics and Law, Wuhan 430073, China; 4School of Economics and Management, Jiangxi Agricultural University, Nanchang 330045, China

**Keywords:** human health, air pollution, thermal inversion, China

## Abstract

Under the background of the far-reaching impact of the COVID-19 epidemic on global economic development, the interactive effect of economic recovery and pollution rebound makes the research topic of air pollution and human health receive attention again. Matching a series of new datasets and employing thermal inversion as an instrumental variable, this study investigates the physical and mental health effect of air pollution jointly in China. We find that in the short run, the above inference holds for both physical and mental health. These short-run influences are credible after a series of robustness checks and vary with different individual characteristics and geographical locations. We also find that in the long run, air pollution only damages mental health. Finally, this study calculates the health cost of air pollution. The above findings indicate that in China, the effect of air pollution on physical and mental health cannot be ignored. The government needs to consider the heterogeneity and long-run and short-run differences in the health effects of air pollution when formulating corresponding environmental and medical policies.

## 1. Introduction

The health effect of air pollution is an enduring research topic. Especially under the background of the far-reaching impact of the COVID-19 epidemic on global economic development, the interactive effect of economic recovery and pollution rebound makes this research topic receive attention again. Numerous studies have come to the fore on the health effects of air pollution [1,2,3,4,5,6,7]. Some literature mainly focuses on whether the rise of air pollution concentration harms physical health, while the other focuses on mental health. However, they only focus on a single perspective and do not examine the influence jointly. Understanding this causal connection and its related health cost from an overall view is highly crucial in that the health damage of individuals exposed to air pollution should be diverse, that is, physical and mental coexistence, rather than unilateral. Besides, governments in various countries have been deeply concerned about the social welfare of individual health.

Another research direction on the health effects of air pollution is to explore the causal effect in the long run. Although this has become a fact of many researches in the medical field, there is still little literature using economic methods to study it. Zhang et al. [1] investigate the long-run mental health effect of air pollution for two years, which provides ideas for further exploration because their study is not based on credible causal inference methods. What makes the long-run effect of air pollution worth our investigation? On the one hand, it may give rise to an increase in the hidden treatment cost when air pollution causes long-term and persistent damage to human health. On the other hand, this influence is an essential factor for the government in formulating environmental policies. If we ignore the long-run effect, the government may underestimate this negative influence and adopt some short-run policies leading to a high level of air pollution concentration for a long time. Therefore, we employ an in-depth investigation of the long-run effect on physical health and mental health jointly.

Moreover, previous related studies primarily focus on areas with low air pollution concentration, which may lead to biased estimation results. A batch of the literature on China suffering severe air pollution opens ideas as the background for our research. The average PM_2.5_ concentration of China during 2010–2018 calculated in our study is 72.44 μg/m^3^, and the maximum value reaches 141.6 μg/m^3^. This abnormally high air pollution level is about seven times higher than the 10 μg/m^3^ proposed by the World Health Organization, a specialized agency under the United Nations and the largest international public health organization, indicating that air pollution in China is severe. China also realizes this severe problem and put forward strict requirements by implementing the healthy China strategy in 2017. Therefore, from the perspective of reality and sample, China presents us with an excellent sample and opportunity to delve deeper into the causal connection between air pollution and human health.

Under the above framework, we further discuss the heterogeneity of the short-run effect of air pollution on physical and mental health, respectively. These heterogeneous effects are guided by a thought that individuals may be affected by different levels of air pollution concentration. Hence, the short-run influence of air pollution may vary with different individual characteristics and geographical locations. That is, people with different lifestyles and jobs experience different levels of air pollution, which will lead to the amplification or reduction of this influence, thus resulting in heterogeneous effects.

On the whole, our work provides new evidence on the human health effects of air pollution and constitutes a novel step toward the research direction of physical and mental health jointly. Roughly speaking, the analyses proceed in three stages. The short-run influence of air pollution on human health is first investigated employing an instrumental variable (IV) approach. Then, a long difference (LD) model is added to explore the long-run influence. Finally, a back-of-the-envelope analysis is applied to roughly assess the health cost of air pollution.

Several different but complementary strategies are implemented to explore the influence of air pollution. First, a detailed set of individual characteristics and weather factors are controlled, including age, education, marital status, income level, social status, household register, employment, insurance, humidity, temperature, and precipitation. These controls filter out individual differences and uncertain weather factors that may affect air pollution and human health together as much as possible. Second, we employ thermal inversion as an IV to accurately estimate the influence of air pollution on human health in the short run. According to this literature [2,8,9,10,11,12], thermal inversion is plausibly exogenous to error terms in the model because artificially creating thermal inversion is temporarily impossible. Third, several robustness checks are used to confirm the validity of IV results, such as replacing the core independent variable, replacing the IV, and the falsification test. Last, we employ an LD model to investigate the long-run influence of air pollution by combining the IV approach.

Drawing on the latest research [6,10,12,13], we combine several datasets. First, the data of individual health and individual characteristics are collected from the China Family Panel Survey (CFPS), which helps measure health conditions at the individual level and allows us to control relevant confounding variables during estimation accurately. Second, the data of air pollution and thermal inversion are collected from the National Aeronautics and Space Administration (NASA) USA. These variables are usually the same for individuals in a region. Therefore, after matching with CFPS data, determining the basic unit of these data to the county can set the data as accurate as possible to the microscopic level, but not so scattered as to deviate from reality. Third, we obtain the data of weather factors from the China National Meteorological Information Center, which is also matched with CFPS data and accurate at the county level.

Based on the above data sets, how to accurately measure individual health and air pollution are the premise for us to implement our estimation strategy. For individual health, we measure physical, mental, and human health, respectively. Physical health is measured by respondents’ rating of their health conditions, with a scale of 1 (very unhealthy) to 5 (very healthy). Mental health is measured by the average of two indicators. One is the respondents’ frequency of depression in a week, with a scale of 1 (more than five days) to 4 (less than one day). The other is respondents’ satisfaction with life, with a scale of 1 (feeling very discontented) to 5 (feeling very contented). These two indicators are used to characterize mental health comprehensively. We then use the average of physical and mental health to measure human health and adjust their weights in the robustness check section. For air pollution, we measure air pollution by PM_2.5_ concentration and SO_2_ concentration. The distribution of these data is calculated through satellite-based aerosol optical depth (AOD) retrieval. The fundamental element of this calculation is that AOD is the optical thickness of aerosol components due to extinction, which is used to describe the attenuation degree of aerosol to light, also called atmospheric turbidity. Its mathematical meaning is the integration of extinction coefficient in the vertical direction, so AOD can reflect the influence of aerosol on radiation balance and is highly related to the near-ground particulate matter. See Section 3 for the data integration process. We finally obtain the data of PM_2.5_ and SO_2_ concentrations in each grid across all Chinese areas from 2010 to 2018 and add them up to the year and county level.

The findings can be summarized in the following manner. First, in the short run, air pollution causes harm to physical and mental health jointly. Specifically, a one standard-deviation (SD) increase in the PM_2.5_ concentration is associated with a 2.648 SD decrease in physical health, a 1.994 SD decrease in mental health, and a 3.683 SD decrease in human health. Second, this negative influence varies with different individual characteristics (age and education) and geographical locations (coast and household register). Third, in the long run (six years and eight years), the rise of air pollution concentration only damages mental health. Fourth, from a back-of-the-envelope analysis, the health cost of air pollution is that a one μg/m^3^ increase in the PM_2.5_ concentration is associated with a 10.98-billion-yuan cost in human health.

Compared with other literature, our contributions lie in four facets. First, we identify the health effect of air pollution by the IV approach, which can solve the endogenous problem. Second, comparing with the previous study in a single direction, we examine the influence of air pollution on physical and mental health jointly. Third, we investigate not only the short-run effect, but also the long-run effect through the LD model. Fourth, our microdata samples are at an individual and county level and can reveal the real health effect of air pollution more than macro data.

## 2. Literature Review

We review the literature mainly from four aspects. First, the effect of air pollution on human health has always been a topic of particular concern in the medical field. Concerning physical health, exposure to air pollution indisputably causes respiratory diseases [14]. Gilmour et al. [15] find that human immune function will be weakened if exposed to airborne particulates. de Oliveira Alves et al. [16] reveal that exposure to air pollution will strongly increase the levels of inflammatory cytokines and DNA damage. Heyes and Zhu [17] discuss that being exposed to air pollution can give rise to sleep quality decline or insomnia caused by shortness of breath, increased heart rate, and upper respiratory tract irritation. In addition, air pollution is likely to cause high risks of cardiac dysfunction [18], cellular immune disorders [19], vascular diseases [20], epigenetic modification [21], and cancer [22]. Concerning mental health, some literature proves that air pollution induces systemic or brain-based oxidative stress and inflammation [23,24,25], which severely damaging cytokine signaling [26]. Besides, staying indoors for a long time will increase people’s anxiety and loneliness [27]. Wagner et al. [28] claim that underlying autonomic neuropathies caused by metabolic disorders may promote inappropriate cardio-regulatory responses to repeated exposure to air pollutants. Fonken et al. [29] find that long-term exposure to air pollutants may lead to emotional behavior changes and cognitive decline. In addition, air pollution can severely damage cytokine signals that regulate brain function, leading to depression [30] and anxiety [31]. The above literature lays a solid foundation for us to study the health effects of air pollution from the economic level.

Second, our study adds to the extensive literature about the effect of air pollution on physical health. These empirical studies can be traced back to around the year 2000 [32,33]. Their estimation results may be endogenous as they mainly focus on time series analysis, or do not control weather factors. In recent years, quite a few studies have provided new evidence on the influence of air pollution on physical health using an IV approach to eliminate endogeneity [4,5,34]. What they have in common is that they measure physical health by hospitalizations for urgent cases of respiratory diseases, hospital admissions for respiratory conditions, and deaths from respiratory diseases. We go further and use respondents’ ratings of their health conditions to measure physical health accurately. Besides, different from their choice of IV, we employ thermal inversion as a brand-new attempt to estimate the influence of air pollution on physical health.

Third, our research contributes to some growing literature about the effect of air pollution on mental health. Part of the studies provides elaborate evidence from diversified perspectives [1,3,35]. Although their works are consummate, the latest part of the studies is more credible because the implementation of the IV approach using thermal inversion. Chen et al. [2] and Zhang et al. [6] argue that increasing PM_2.5_ concentration will induce individual mental illness and reduce their subjective well-being level. In the long run, air pollution may reduce corporate performance and investment [36], which will in turn cause workers to worry about unemployment and reduced income, and adversely affect workers’ mental health [37]. Most of these studies use respondents’ ratings of their mental status to measure mental health based on the latest micro-survey data, and they also use thermal inversion as an IV. These cutting-edge attempts help us to further study the mental health effects of air pollution.

Fourth, our research methodologically relates to many recent works applying thermal inversion as an IV to investigate the influence of air pollution. These works pay attention to infant mortality [8], road safety [9], obesity [10], and manufacturing firm productivity [11], whereas we focus on human health. Besides, our study relates broadly to much previous literature using an LD model, which relies on variables changing slowly or over an extended period [38,39,40]. We apply the LD model to investigate the long-run influence (six years and eight years) of air pollution on human health, which fills the gap in this research direction.

The most relevant literature to our study is Shen et al. [41], which examines the physical and mental health effects of air pollution jointly. Our improvements include three points. The first is that they use the data from the China Health and Retirement Longitudinal Study, focusing on middle-aged and elderly individuals. Our sample includes individuals aged 16 to 94, which is more comprehensive. The second is that all their estimation processes do not use the IV approach, so the estimation results are likely to be biased and unreliable. We apply thermal inversion as an IV to eliminate endogeneity. The third is that they only focus on short-run effects, while we explore the short-run and long-run impact together.

## 3. Estimation Strategy and Data

### 3.1. Estimation Strategy

We define individuals as the basic unit of research, and the main target is to accurately identify the marginal influence of air pollution on human health. This unbiased estimation is conditional on the fact that independent variables are uncorrelated with the error term—the conditional mean independence assumption. That is, the error term, which represents unobserved factors other than independent variables, has an expected value of zero given any value of the independent variables. Although appropriate control variables can be added to the model to alleviate the problem of omitted variables and satisfy the conditional mean independence assumption, estimation bias may still be caused by the following reasons.

First, the source of air pollutants is a large part of the production activities of enterprises, so it is usually related to many mixed economic factors, such as labor productivity and commodity price. These confounding factors are difficult to observe and measure and have an uncertain influence on human health. Therefore, due to the unavailability of data, we cannot control these factors in the model, which leads to estimation bias. Second, the policy factors affecting medical systems, and then affecting human health are different from place to place. During our research period, the medical systems in various areas of China have experienced significant changes. The local governments of provinces, cities, and counties have different enforcement levels and ways for the policies issued by the central government. Some areas may have been at a high medical level, whereas the medical level in other places may be undergoing great changes. This kind of policy difference in various areas is difficult to measure, which leads to estimation bias. Third, there may be a reverse causality between human health and air pollution. The increase in air pollution concentration will harm physical and mental health and reduce individual labor time and labor efficiency. In production activities, the reduction of labor input and capacity will further reduce the emission of air pollutants, which leads to estimation bias.

Therefore, we employ thermal inversion as an IV to solve the problem of estimation bias and set the two-stage least squares (2SLS) model as follows.
(1)$$H_{ict}=a_{0}+a_{1}P_{ct}+a_{2}X_{ict}+a_{3}W_{ct}+\lambda_{i}+\theta_{t}+\varepsilon_{ict}$$
(2)$$P_{ct}=\beta_{0}+\beta_{1}T_{ct}+\beta_{2}X_{ict}+\beta_{3}W_{ct}+\lambda_{i}+\theta_{t}+\mu_{ict}$$
where dependent variable $H_{ict}$ is an outcome (physical health, measured by the respondent’s answer to the question of health conditions; mental health, measured by the respondent’s answer to the question of depressed frequency and satisfaction with life; and human health, measured by the average of physical and mental health) of individual i in county c in year t; independent variable $P_{ct}$ denotes air pollution in county c in year t, measured by the PM_2.5_ concentration and SO_2_ concentration (we redefine the measurement index of air pollution in Section 3). IV $T_{ct}$ denotes thermal inversion in county c in year t, measured by the number of days in a year when inversion occurs; $X_{ict}$ are time-varying individual-level controls, namely, age, education, marital status, income level, social status, household register, employment, and insurance, intended to capture time-varying individual factors; $W_{ct}$ are time-varying county-level controls, namely, humidity, temperature, and precipitation, intended to capture time-varying weather factors; $\lambda_{i}$ is an individual fixed effects term capturing time-invariant individual characteristics such as gender, weight, and geographic locations; $\theta_{t}$ is a year fixed effects term capturing time-invariant macro shocks common to all individuals such as the Healthy China Strategy implemented by the Chinese government; $\varepsilon_{ict}$ and $\mu_{ict}$ are the error terms.

The IV estimates rely on two key identification assumptions: the first is that thermal inversion is related to air pollution, which can be tested by the first-stage regression. The second is that thermal inversion is not related to the error term. That is, the distribution of thermal inversion is quasi-random. This is also known as the exclusive hypothesis, which means that thermal inversion does not affect individual health through other channels except air pollution. We report the KP rk Wald F statistic in Section 3 to resolve this doubt.

### 3.2. Data

*CFPS Data*—The data of health conditions and individual characteristics in this study come primarily from CFPS, one of the longest longitudinal and most comprehensive health surveys in China, which is still in progress. Through long-term tracking and collection, it reflects the changes in essential variables such as society, economy, population, education, and health in China. Since the official start of the survey in 2010, follow-up surveys have been performed every two years, and the latest data are up to 2018. The survey covers about 15,000 households and more than 40,000 people in 25 provinces of China. It uses the method of systematic probability sampling, which is multistage, implicit stratification, and probability proportional to size. Most importantly, CFPS data have three advantages compared with other data. First, it takes individuals as the basic unit and provides the survey results of individual health, which is conducive to studying the effect of air pollution on physical and mental health jointly. Second, it covers a series of personal information of respondents, enabling the accurate control of relevant factors at the individual level in the model. Compared with adding the individual-year fixed effect in the model, it can keep more degrees of freedom. Last, it provides the geographic location of every respondent, which enables us to precisely match individual-level data (human health and individual characteristics) to county-level data (air pollution, thermal inversion, and weather factors).

The following are the details of the above CFPS data. First, physical health is measured by respondents’ rating of their health conditions, with a scale of 1 (very unhealthy) to 5 (very healthy). Mental health is measured by the average of two indicators. One is the respondents’ frequency of depression in a week, with a scale of 1 (more than five days) to 4 (less than one day). The other is respondents’ satisfaction with life, with a scale of 1 (feeling very discontented) to 5 (feeling very contented). These two indicators are used to characterize mental health comprehensively. We then use the average of physical and mental health to measure human health and adjust their weights in the robustness check section. Our measurement of physical and mental health is reasonable to some extent. As for physical health, the existing literature mainly use the hospitalization rate of respiratory diseases or mortality rate of respiratory diseases to measure it. Although these indicators are closer to reality, as mentioned in literature review section, the literature in medical field proves that air pollution has far more effects on physical health than respiratory diseases. Therefore, the indicators based on the survey data can measure the physical health more comprehensively. As for mental health, mental health is a subjective concept. Much recent and relevant literature uses different variables to measure it. What we try to do is to use micro-data to measure it as accurately as possible, which is the main form of study by economists. Second, individual characteristics include age, education, marital status, income level, social status, household register, employment, and insurance. Among them, besides age and education, other variables are either the individuals’ self-rating, ranging from 1 to 5, or a dummy variable. A detailed description of these individual characteristics will be reported later.

Air Pollution and Thermal Inversion Data—The data of air pollution and thermal inversion come from NASA USA. We measure air pollution by PM_2.5_ concentration and SO_2_ concentration. According to Zhang et al. [6], these data obtained from the National Environmental Monitoring Center of China’s Ministry of Environmental Protection, which is used in much of the existing literature, have some shortcomings, such as high monitoring cost, which leads to the low frequency of observation data, few monitoring stations, which leads to a limited number of samples, uneven distribution, which leads to a biased distribution of observed data, and artificial tampering, which leads to the false data we get. To avoid such disadvantages, we use satellite-based AOD retrieval to calculate the distribution of PM_2.5_ concentration and SO_2_ concentration. The fundamental of calculation is that AOD is the optical thickness of aerosol components due to extinction, which is used to describe the attenuation degree of aerosol to light, also called atmospheric turbidity. Its mathematical meaning is the integration of the extinction coefficient in the vertical direction, so AOD can reflect the influence of aerosol on radiation balance and is highly related to the near-ground particulate matter. Referring to Buchard et al. [13] and Chen et al. [12], we obtain the grid data of AOD based on longitude and latitude (0.5° × 0.625°, around 50 km × 60 km) since 1980 from NASA. Then, we use Arcgis10.2 software to obtain the data of PM_2.5_ and SO_2_ concentrations in each grid across all Chinese areas from 2010 to 2018 and add them up to the year and county level.

Furthermore, we employ thermal inversion as an IV. The data of thermal inversion present the average temperature every 6 h in a grid of 50 km × 60 km per 42 atmospheric layers above sea level since 1980. In every six hours, we add up from the grid to the county level for each year and each layer, then obtain the preliminarily processed data of thermal inversion. What makes thermal inversion a good IV for air pollution? The phenomenon of thermal inversion has been widely proven by scholars that it is not deliberately caused by human beings [6,8,10,11,12]. According to the academic definition, the higher the altitude, the lower the temperature, whereas in specific weather circumstances, there is a phenomenon that the temperature increases with the altitude, which is called “thermal inversion” in meteorology. Thermal inversion hinders the vertical convection of air and the diffusion of pollutants and then causes the aggravation of air pollution near the ground. Under the current technical conditions, artificially creating thermal inversion is temporarily impossible. Therefore, thermal inversion proves of great use to explore whether the rise of air pollution concentration harms human health. We use the preliminarily processed data reporting the average temperature every 6 h to construct the thermal inversion variable. We define the times that thermal inversion happens in one day and it can be judged whether thermal inversion occurs four times every day. To formulate a consistent criterion for judgment and decrease measurement error as much as possible, we define that thermal inversion happens in a county in one day if thermal inversion occurs at least once on that day. Therefore, thermal inversion occurs when the temperature of the layer of 110 m is lower than that of the layer of 320 m according to the temperature difference between these two layers of the atmosphere [9,10,11]. Finally, we add up the thermal inversion data in the time dimension to the year level, and in the unit dimension to the county level, and then match it with air pollution data.

Weather Data—The weather data come from the National Meteorological Information Center of China. More than 800 weather stations in this center provide daily weather data, including temperature, precipitation, relative humidity, etc. Different from related literature [1,6,42], we control humidity, temperature, and precipitation in the model, which may influence human health and air pollution, but are not strongly related to thermal inversion. Some variables, such as sunshine duration, may be multicollinear with air pollution and temperature, although they can affect individuals’ emotions and social behaviors. Therefore, adding these variables to the model may absorb the variation of air pollution, leading to the bias of estimated coefficients. For the processing of these weather factors, we select a radius of 200 km and employ an inverse-distance weighting approach to convert the data obtained from the station to the county level [10].

Table 1 reports the summary statistics of main variables. Note that air pollution may lead to population migration and cause estimation bias. Thus, we compare the migration of each family in the CFPS data used in this study. After comparison, 99.71% of individuals in the sample reside in the same county, so we can consider that no population migration exists. Table 1 presents that the average physical, mental, and human health is 2.933, 3.506, and 3.217, respectively. This suggests that after comparing with the maximum and minimum values, individuals’ health conditions are generally at a medium level. The average PM_2.5_ concentration is 72.44 μg/m^3^, and the maximum value reaches 141.6 μg/m^3^. This abnormally high air pollution level is about seven times higher than the 10 μg/m^3^ proposed by the World Health Organization, a specialized agency under the United Nations and the largest international public health organization, indicating that air pollution in China is severe. The average thermal inversion is 158, suggesting that inversion occurs for nearly half a year.

## 4. Results

### 4.1. Ordinary Least Squares (OLS) Estimates

First, OLS estimates are used to preliminarily investigate whether air pollution cause harms to physical, mental, and human health. Knowing that many confounding variables are unobservable and uncontrollable, we only use OLS estimation results for comparison. Then, IV estimates are used to eliminate endogeneity to further investigate this influence accurately. Through the comparison between them, the causal connection between air pollution and human health can be understood intuitively.

Panels A and B in Table 2 report the results from the OLS estimates in Equation (1) with and without controlling for individual characteristics and weather factors (Panel A in Appendix A Table A1 reports the coefficients on all variables). Columns (1), (2), and (3) represent physical, mental, and human health, respectively. In Column (1), the estimated coefficients of physical health are −0.0040 and −0.0036. They are consistently negative and statistically significant at the 1% level, suggesting that air pollution may harm individuals’ physical health. In Column (2), the estimated coefficients of mental health are 0.0024 and 0.0019. They are consistently positive and statistically significant at the 1% level and 5% level, indicating that air pollution may have a positive influence on individuals’ mental health. In Column (3), the estimated coefficients of human health are −0.0007 and −0.0008. They are close to 0 and statistically insignificant, suggesting that it may be open to question whether air pollution affects human health from the overall perspective. The above results are different from our theoretical and practical expectations, which may be due to the estimation bias previously discussed. Therefore, further using the IV approach is necessary to solve the endogenous problem and to investigate this influence accurately.

### 4.2. IV Estimates

Table 3 reports the estimates from the 2SLS regressions in Equations (1) and (2) with controlling for individual characteristics and weather factors (Panel B in Appendix A Table A1 reports the coefficients on all variables). The first-stage results show that the estimated coefficients of PM_2.5_ are all statistically significant at the 1% level. In addition, the LM statistics of the unidentifiable test are all significant at the 1% level, suggesting that the rank condition holds. Moreover, the F statistics of the weak IV test are all greater than 16.38 (10% maximal IV size), suggesting that thermal inversion is not a weak IV. Therefore, the above results together show that thermal inversion is correlated to PM_2.5_.

The second-stage results reveal that the estimated coefficients of physical, mental, and human health are all negative and statistically significant at the 1% and 5% level. Therefore, we can judge that, after solving the endogenous problem using the IV approach, air pollution causes harm to physical and mental health, which is different from the results of OLS estimates. Specifically, the estimated coefficient of physical health is −0.1147, that is, a one SD increase in the PM_2.5_ concentration (ug/m^3^) is associated with a 2.648 SD decrease in physical health. The estimated coefficient of mental health is −0.0450, that is, a one SD increase in the PM_2.5_ concentration (ug/m^3^) is associated with a 1.994 SD decrease in mental health. The estimated coefficient of human health is −0.0797, which means that a one SD increase in the PM_2.5_ concentration (ug/m^3^) is associated with a 3.683 SD decrease in human health. Through the comparison of the above results, we can roughly draw a conclusion that air pollution may cause more harm to physical health than to mental health.

We put forward a possible explanation on these estimation results. The basic time unit of our data is the year, so the above estimation results are the average annual effect. From the point of view of physical health, usually within one year, individuals are likely to suffer from respiratory diseases due to air pollution. It always takes several months to realize that they are sick, go to the hospital for diagnosis and then take treatment. From the point of view of mental health, when faced with a living environment full of smog, blocked vision, and peculiar smell in the air, people may not be mentally sick in the short term, but only feel a little uncomfortable.

### 4.3. Robustness Check

The results of IV estimates are robust to various alternative variables, adjusted weights, and the falsification test. First, we employ the PM_2.5_ to denote air pollution in the benchmark regression, while air pollution may also be caused by other air pollutants such as SO_2_. In this section, the SO_2_ concentration is thus used to denote air pollution to enhance the robustness of the benchmark regression, and we investigate this influence directly under the framework of IV estimation. Table 4 reports the results from the IV estimates by replacing the core independent variable.

As presented in Table 4, the first-stage results show that the estimated coefficients of SO_2_ are all statistically significant at 1% level, together with the results of LM and Wald F statistics, suggesting that thermal inversion is correlated to SO_2_. The second-stage results reveal that the estimated coefficients of physical, mental, and human health are all negative and statistically significant at the 1% and 5% level. Thus, after replacing the core independent variable, air pollution still has a negative influence on physical and mental health. Specifically, the estimated coefficient of physical health is −0.2461, that is, a one SD increase in the SO_2_ concentration (ug/m^3^) is associated with a 2.755 SD decrease in physical health. The estimated coefficient of mental health is −0.0963, that is, a one SD increase in the SO_2_ concentration (ug/m^3^) is associated with a 2.069 SD decrease in mental health. The estimated coefficient of human health is −0.1703, that is, a one SD increase in the SO_2_ concentration (ug/m^3^) is associated with a 3.816 SD decrease in human health. Through the comparison of the above results with the one in Panel C in Table 2, we can come to the conclusion that whether the PM_2.5_ concentration or the SO_2_ concentration is used to measure air pollution, the results of IV estimates are consistent.

Second, we adjust the definition of thermal inversion to test the rationality of the IV. Previously, we judged whether thermal inversion, the phenomenon that temperature increases with altitude, happens by the temperature difference between the layer of 110 m and the layer of 320 m of the atmosphere: when the temperature of the layer of 110 m is lower than that of the layer of 320 m, it is considered that thermal inversion occurs on this day. In this section, we redefine thermal inversion based on the temperature difference between the layer of 110 m and the layer of 540 m of the atmosphere: when the temperature of the layer of 110 m is lower than that of the layer of 540 m, it is considered that thermal inversion occurs on this day [6]. Table 5 reports the results from the IV estimates by replacing the IV.

As shown in Table 5, the first-stage results reveal that the estimated coefficients of PM_2.5_ are all statistically significant at the 1% level, together with the results of LM and Wald F statistics, suggesting that thermal inversion is correlated to PM_2.5_. The second-stage results show that the estimated coefficients of physical, mental, and human health are all negative and statistically significant at the 1% level and 5% level. This means that, after replacing the IV, air pollution still causes harm to physical and mental health, which is the same as the benchmark regression. Therefore, the benchmark regression results are robust.

Third, we define human health as the average of physical health and mental health in the benchmark regression. However, in an actual situation, the health conditions of human beings are not as clearly distinguished as it is in this study, which means that the ratio of physical health to mental health may be different. Thus, we readjust the ratio to define human health in the ratios of 3:7, 4:6, 6:4, and 7:3. Table 6 reports the results from the IV estimates by readjusting the ratio of physical health to mental health.

As displayed in Table 6, the results are the same as the benchmark regression. The estimated coefficients of different human health are all negative and statistically significant at 1% level. Therefore, air pollution still causes harm to human health after readjusting the ratio of physical health to mental health. These results further confirm the robustness of the benchmark regression.

Last, a falsification test is proposed to illustrate the robustness of the benchmark regression from the opposite side. The main idea of the falsification test is to use a variable unrelated to air pollution as a dependent variable for regression. Theoretically, there should be no causal connection between them. Therefore, when the estimated coefficient is insignificant, the benchmark regression results are proven to be robust. We use individuals’ height as a dependent variable, which should not correlate with air pollution. Table 7 reports the results from the IV estimates with a falsification test.

As expected, all the estimated coefficients in Table 7 are statistically insignificant whether using the PM_2.5_ concentration or the SO_2_ concentration to measure air pollution, suggesting that the rise of air pollution concentration is harmless to individuals’ height. Hence, our falsification strategy is reasonable and valid.

## 5. Heterogeneous Effects

Having proven a robust connection between air pollution and human health, we now turn to explore how this negative influence vary with different individual characteristics and geographical locations. These heterogeneous effects are guided by a thought that individuals may be affected by different levels of air pollution concentration. Hence, the short-run influence of air pollution may vary with different individual characteristics and geographical locations. That is, people with different lifestyles and jobs experience different levels of air pollution, which will lead to the amplification or reduction of this influence, thus resulting in heterogeneous effects.

First, for individual characteristics, there may be heterogeneous effects with age and education on account of the differences in workplace and life form between these groups. Therefore, we divide the samples into the group of the young (≤40), the middle (40–60), and the old (>60) by age, and the group of those who have not completed or just completed compulsory education (≤9 years) and those who have completed compulsory education (>9 years) by education. The estimation results of the different influence among individual characteristics are reported in Table 8 and Table 9.

Table 8 shows the heterogeneity effects of air pollution on physical and mental health with age. In Columns (21) and (23), the estimated coefficients of physical health are negative and statistically significant for the young (≤40) and the middle (40–60). Besides, although the estimated coefficients of mental health are negative, they are not statistically significant for the young (≤40) and the middle (40–60). In columns (25) and (26), the estimated coefficients of physical and mental health are statistically insignificant for the old (>60). Therefore, we can know that air pollution affects physical health mainly for the young (≤40) and the middle (40–60), not the old (>60). The reason may be that the young ones go outside to participate in work and activities more often compared with the old. Hence, the young have more opportunities and longer exposure to air pollution, which causes more harm to their physical health.

Table 9 shows the heterogeneity effects of air pollution on physical and mental health with education. In Columns (27) and (28), the estimated coefficients of physical and mental health are negative and statistically significant for those who have not completed or just completed compulsory education (≤9 years). In contrast, the estimated coefficients of physical and mental health in Columns (29) and (30) are statistically insignificant for those who have completed compulsory education (>9 years). Therefore, air pollution causes harm to physical and mental health mainly for those who have not completed or just completed compulsory education (≤9 years), not those who have completed compulsory education (>9 years). The reason may be that the shorter the education year, the more likely it is for people to participate in outdoor physical labor. In contrast, well-educated people may work indoors, such as doctors, lawyers, and government officials. Hence less educated people are exposed to air pollution for a longer time, which causes more harm to their physical and mental health.

Second, for geographical locations, there may be heterogeneous effects with the coast and household register because the differences in economic factors causing different degrees of air pollution between these groups. Therefore, we divide the samples into the group of coastal areas and inland areas by coast, and the group of urban areas and rural areas by the household register. The estimation results of the different influences between geographical locations are reported in Table 10 and Table 11.

Table 10 shows the heterogeneity effects of air pollution on physical and mental health with the coast. In Columns (31) and (32), the estimated coefficients of physical and mental health are negative and statistically significant for coastal areas, suggesting that in coastal areas, air pollution harms both physical and mental health. In Column (33), the estimated coefficient of physical health is negative and statistically significant, whereas, in Column (34), the estimated coefficient of mental health is statistically insignificant. Therefore, in inland areas, air pollution only harms physical health. The inland area versus coastal area difference is positive and statistically significant, suggesting a difference in physical health between coastal and inland areas. It may be because the economic development level in the coastal areas of China is higher than that in inland areas, with a higher degree of industrialization, so air pollution is also more severe. Hence, the people in coastal areas are exposed to high-concentration air pollution, which causes harm to their physical and mental health. Our data also proves this inference, the mean value of the PM_2.5_ concentration in inland areas is 68.72 ug/m^3^, and 77.36 ug/m^3^ in coastal areas.

Table 11 presents the heterogeneity effects of air pollution on physical and mental health with household register. In Column (35), the estimated coefficient of physical health is negative and statistically significant, whereas in Column (36), the estimated coefficient of mental health is statistically insignificant. Therefore, in urban areas, air pollution only harms physical health. In Columns (37) and (38), the estimated coefficients of physical and mental health are negative and statistically significant for rural areas, indicating that in rural areas, air pollution harms physical and mental health. The urban area versus rural area difference is positive and statistically significant, suggesting a difference in physical health between urban and rural areas. The reason may be that there is a big gap between urban and rural medical conditions in China. Huang and Wu [43] made three proofs of this inference. First, urban and rural residents participate in different medical insurance programs, and urban residents can enjoy more benefits. Second, the rural health insurance programs were administered at a low level (e.g., county), which greatly weakens the risk sharing and portability of health insurance. Third, the urban and rural health insurance programs were administered by different ministerial bodies and line bureaucracies. Information and resource for health insurance were rarely shared between them. Hence, people in rural areas may not be able to be treated immediately and effectively for diseases caused by air pollution.

## 6. Long-Run Influence

The analysis thus far pays close attention to the short-run influence of air pollution. We now focus on the long run and conduct an in-depth study. The research on the long-run effects of air pollution is significant. On the one hand, it may give rise to an increase in the hidden treatment cost when air pollution causes long-run and persistent damage to human health. On the other hand, this influence is an essential factor for the government in formulating environmental policies. If we ignore the long-run effect, the government may underestimate this negative influence and adopt some short-run policies leading to a high level of air pollution concentration for a long time. We use the LD model to estimate the long-run influence. LD model relies on variables changing slowly (such as temperature changes, aging population, etc.) or with an extended period (such as a period of several years) and is essentially a first-order difference for all variables. The specific LD model is set as follows.
(3)$$H_{ict}-H_{ict-k}={\tilde{\alpha }}_{1}\left(P_{ct}-P_{ct-k}\right)+{\tilde{\alpha }}_{2}\left(X_{ict}-X_{ict-k}\right)+{\tilde{\alpha }}_{3}\left(W_{ct}-W_{ct-k}\right)+{\tilde{\theta }}_{t}+{\tilde{\varepsilon }}_{ict}$$
where the dependent variable $H_{ict}-H_{ict-k}$ is the change in outcome (physical health, mental health, and human health over time), which reflects their accumulation between k years; other independent variables are defined analogously, but note that in this model, the individual fixed effects are eliminated through the difference; ${\tilde{\theta }}_{t}$ is a year fixed effects term and ${\tilde{\varepsilon }}_{ict}$ is the error term.

The estimation results from Equation (3) may be endogenous because of the reasons discussed in Section 2. Hence, we set the following 2SLS model on the basis of the LD model using thermal inversion in year t as an IV.
(4)$$P_{ct}-P_{ct-k}={\tilde{\beta }}_{0}+{\tilde{\beta }}_{1}T_{ct}+{\tilde{\beta }}_{2}\left(X_{ict}-X_{ict-k}\right)+{\tilde{\beta }}_{3}\left(W_{ct}-W_{ct-k}\right)+{\tilde{\theta }}_{t}+{\tilde{\mu }}_{ict}$$

The settings of Equation (4) are the same as before, and all variables are defined uniformly from Equation (3). IV $T_{ct}$ denotes thermal inversion in the current year. By combining the LD model and the 2SLS model, the new model from Equations (3) and (4) can capture a wider range of influence of air pollution on human health. Specifically, we define k = 6 to investigate the long-run influence over 2010–2016 and 2012–2018 and define k = 8 to investigate the long-run influence over 2010–2018. The estimation results are reported in Table 12 and Table 13.

As reported in Table 12, the first-stage results show that the estimated coefficients of ΔPM_2.5_ concentration are all statistically significant at the 1% level, together with the results of LM and Wald F statistics, suggesting that thermal inversion in the current year is correlated to ΔPM_2.5_ concentration. The second-stage results reveal that the estimated coefficients of mental and human health are negative and statistically significant at the 1% level, whereas the estimated coefficient of physical health is statistically insignificant. Hence, in the six-year-long run, air pollution mainly causes harm to mental health, not physical health.

As displayed in Table 13, the first-stage results reveal that the estimated coefficients of ΔPM_2.5_ concentration are all statistically significant at the 1% level, together with the results of LM and Wald F statistics, suggesting that thermal inversion in the current year is correlated to ΔPM_2.5_ concentration. The second-stage results show that the estimated coefficients of mental and human health are negative and statistically significant at 1% level. In contrast, the estimated coefficient of physical health is statistically insignificant. Hence, in the eight-year-long run, air pollution mainly causes harm to mental health, not physical health.

From the estimation results in Table 12 and Table 13, we can draw a conclusion that in the long run, air pollution only causes harm to individuals’ mental health, but not to their physical health. This conclusion seems to be contrary to our common sense. However, our estimation approaches have eliminated most endogeneity, and the estimation results have been compared between the six-year and eight-year differences, so we hold that this conclusion is robust. We further propose two possible reasons why the long-run influence of air pollution on human health mainly comes from a mental perspective. The first is that in the long run, the damage to individuals’ physical health caused by air pollution can be treated, such as bronchitis, asthma, pulmonary edema, emphysema, dyspnea, and headache. In contrast, the negative influence on individuals’ mental health is challenging to treat. For example, once a person suffers from depression, he may be affected by it all his life. The second is that in the long run, people can predictably prevent damage to their physical health by wearing masks, cleaning their noses and mouth, and using air purifiers. However, during this process, taking these preventive measures for a long time will negatively affect individuals’ mental health. Moreover, the damage to individuals’ mental health caused by air pollution may be invisible. It is complicated to cure them when they realize that their mental condition has deteriorated highly.

## 7. Health Cost of Air Pollution

Whether the rise of air pollution concentration is harmful to human health has been fully investigated by previous analyses, but the aggregate welfare implications of this influence remain unclear. Understanding this causal connection and its related health cost from an overall view is highly crucial in that the health damage of individuals exposed to air pollution should be diverse, that is, physical and mental coexistence, rather than unilateral. Besides, governments in various countries have been deeply concerned about the social welfare of individual health. We investigate this issue by performing a back-of-the-envelope health-cost analysis. The specific equation is set as follows.
(5)Health cost of air pollution=|∂H∂P| /MeanH×Costyear
where ∂H∂P is the average marginal substitution rate between health and air pollution. $Mean_{H}$ is the mean value of physical or mental health. |∂H∂P| /MeanH denotes the percentage decrease in health caused by a one-unit increase in the PM_2.5_ concentration (ug/m^3^). $Cost_{year}$ is the annual cost of treatment for physical or mental health. Thus, (∂H∂P) /MeanH times $Cost_{year}$ equals the health cost of one ug/m^3^ PM_2.5_ concentration.

Concerning physical health, the coefficient in Column (4) in Table 3 is −0.1147, and the mean value of physical health in Table 1 is 2.933. In addition, according to Qi [44], the corresponding annual health cost of patients with respiratory diseases in China is CNY 222.61 billion. Therefore, a one μg/m^3^ increase in the PM_2.5_ concentration is associated with an CNY 8.706 billion cost in physical health.

Concerning mental health, the coefficient in Column (5) in Table 3 is −0.045, and the mean value of physical health in Table 1 is 3.506. In addition, according to Xu et al. [45], the annual cost of depression in China is USD 3665 (22,686 yuan) for individual patients. Approximately 95 million people in China have depression [46], but only 8.2% of patients with depression in China actively seek treatment in hospitals [47]. If this 8.2% of patients are treated, the corresponding annual health cost is about CNY 176.7 billion. Therefore, a one μg/m^3^ increase in the PM_2.5_ concentration is associated with a CNY 2.269 billion cost in mental health.

Combining the calculation of the above physical and mental health costs of air pollution, we can obtain a rough result: a one μg/m^3^ increase in the PM_2.5_ concentration is associated with a CNY 10.98 billion cost in human health. In addition, the PM_2.5_ concentration is 73.43 μg/m^3^ in 2010 and 69.69 μg/m^3^ in 2018, which means that China’s PM_2.5_ concentration decreased by 3.74 μg/m^3^ from 2010 to 2018. Therefore, the total health cost of air pollution from 2010 to 2018 is CNY 41.13 billion (3.74 μg/m^3^ times CNY 10.98 billion), and the average annual health cost of air pollution is CNY 5.14 billion (CNY 41.13 billion divided by 8 years). In 2018, the medical and health expenditure arranged in China’s fiscal budget is CNY 1529 billion. Therefore, our calculated average annual health cost of air pollution account for 0.34% of it (CNY 5.14 billion divided by CNY 1529 billion).

Our results are only preliminary and are rough calculations. Due to the difficulty of accurate estimate, we do not consider the mixed effect of air pollution on physical health and mental health. This effect may be considerable because these two kinds of health damage can influence each other. Therefore, our results may underestimate the health cost of air pollution, which deserves further investigation in other studies.

## 8. Conclusions and Implication

Under the background of economic recovery in the post-COVID-19 epidemic era, understanding the causal connection between human health and air pollution and its related health cost from an overall perspective is highly crucial. Numerous studies only focus on a single view of physical or mental health. They do not examine whether the rise of air pollution concentration is harmful to human health from an overall perspective yet. Besides, much less is known about whether this influence exists in the long run. To fill this gap, this study explores the effects of air pollution on physical and mental health jointly in China at the individual level. Matching a series of new datasets and employing thermal inversion as an IV, we strengthened the credibility of the estimation results.

Through comparing OLS and 2SLS regressions, the findings constitute compelling evidence that the rise of air pollution concentration causes harm to physical, mental, and human health. Specifically, a one SD increase in the PM_2.5_ concentration (ug/m^3^) is associated with a 2.648 SD decrease in physical health, a 1.994 SD decrease in mental health, and a 3.683 SD decrease in human health. These estimation results are credible to several robustness checks. The damage to physical and mental health caused by air pollution is mainly for the young, less educated people, coastal areas, and rural areas. Through the LD estimates, we find that in the long run (six years and eight years), the rise of air pollution concentration harms mental health, but not physical health. From the back-of-the-envelope analysis, we roughly calculate that a one μg/m^3^ increase in the PM_2.5_ concentration is associated with a CNY 10.98 billion cost in human health.

In contrast to other papers, this study delves into what kind of health is influenced by air pollution and whether the influence exists in the long run, which is a further step toward understanding the causal connection between air pollution and human health and related health cost. However, due to the difficulty of obtaining data, more must be accomplished in future research. First, it is necessary to select more appropriate indicators to measure individuals’ physical, mental, and human health. The accurate measurement is central to a full comprehending of the causal connection between air pollution and human health. Second, further efforts should precisely investigate long-run influence by setting a frontier causal inference model. IV estimation is only a method to infer the real effect of air pollution on individual health, and it may be more credible to find a reliable exogenous shock. Third, the same and different mechanisms of long-run and short-run influence need to be studied more clearly. A clear influence channel is a strong support for understanding the causal connection between air pollution and individual health from an economic perspective.

Our findings can yield significant policy implications.

First, we should not only pay attention to the impact of air pollution on physical health, but also on mental health. Controlling air pollution can promote the subjective well-being of all social classes. Therefore, it is necessary to promote the equalization of basic public services focusing on health care and social security, which are equally important in physical and mental health. Besides, the list of diseases caused by air pollutants should be included in the basic medical security, and those who return to poverty due to air pollution should be included in the minimum living security. Moreover, it is necessary to establish a binding minimum standard of environmental quality to meet the basic physical and mental health needs.

Second, the urbanization process of China, which relies heavily on resources and energy consumption, has brought great pressure on the regional environmental capacity, resulting in severe air pollution. On the one hand, it is necessary to speed up the integration of urban and rural development, so that the living environment of rural residents is equally clean, and the living conditions are more convenient. Besides, the government could set isolation zones in industrial production areas and living areas to reduce the impact of industrial production on daily life, thus enhancing the happiness of residents. On the other hand, enterprises should intensify efforts to improve the working environment and provide employees with a clean working environment, which can improve residents’ subjective well-being.

Third, the root of air pollution control lies in changing the mode of economic growth, relying on scientific and technological progress, and promoting the upgrading of industrial structure. Therefore, the central government should speed up pollution control legislation and clarify the responsibility distribution between the government and enterprises in air pollution control. Multi-objective incentive mechanism including environmental governance can be introduced into the performance appraisal of local governments, so that local officials can pay attention to the prevention and control of air pollution while promoting urban economic growth. If air pollution is not treated in time, it will cause huge losses to people’s livelihood. The cost of governance in retrospect will be enormous.

## Figures and Tables

**Table 1 ijerph-20-02385-t001:** Summary statistics.

Variable	Definition	*N*	Mean	Std. Dev.	Min	Max
**Individual level**						
Physical health	Respondents’ ratings of health conditions, with a scale of 1 (very unhealthy) to 5 (very healthy)	63,334	2.933	1.305	1	5
Mental health	Average of respondents’ frequency of depression in a week, with a scale of 1 (more than five days) to 4 (less than one day), and respondents’ satisfaction with life, with a scale of 1 (feeling very discontented) to 5 (feeling very contented)	62,601	3.506	0.704	1	4.500
Human health	Average of physical health and mental health	62,600	3.217	0.675	1	4.750
Age	Age in years	63,339	50.45	13.67	16	94
Education	Years of Education	63,134	6.435	4.741	0	20
Marital status	Marriage status (1 = single, 2 = marriage, 3 = cohabitant, 4 = divorced, 5 = lose a spouse)	63,334	2.148	0.745	1	5
Income level	Satisfaction with income level (1 = very discontented, 2 = discontented, 3 = medium, 4 = contented, 5 = very contented)	59,981	2.491	1.045	1	5
Social status	Scoring of social status (1 = very low status, 2 = low status, 3 = medium, 4 = high status, 5 = very high status)	62,430	2.931	1.051	1	5
Household register	Type of Household register (0 = countryside, 1 = city)	63,008	0.443	0.497	0	1
Employment	Work status (0 = unemployment, 1 = employment)	62,259	0.678	0.467	0	1
Insurance	Whether to buy insurance (0 = no, 1 = yes)	63,340	0.914	0.280	0	1
**County level**						
PM_2.5_	PM_2.5_ concentration (μg/m^3^)	63,340	72.44	31.18	7.292	141.6
SO_2_	SO_2_ concentration (μg/m^3^)	63,340	26.20	15.12	0.816	63.60
Thermal inversion	Annual days with thermal inversions	63,340	158.1	77.69	4	324
Humidity	Relative humidity (%)	62,973	66.00	9.108	42.40	86
Temp	Temperature (°C)	63,083	14.31	4.701	−0.90	23.70
Precipitation	Precipitation (mm)	61,326	946.9	543.0	132.2	3203

**Notes:** The samples of our study include 12,668 individuals from 151 counties in 23 provinces and the time is five years in 2010, 2012, 2014, 2016, and 2018. The definition, observations, mean value, standard deviation, minimum value, and maximum value of individual-level data and county-level data are reported in this table.

**Table 2 ijerph-20-02385-t002:** OLS estimates of the influence of air pollution on physical, mental, and human health.

	Physical Health	Mental Health	Human Health
	(1)	(2)	(3)
**Panel A: OLS**			
PM_2.5_	−0.0040 ***	0.0024 ***	−0.0007
	(0.0013)	(0.0008)	(0.0007)
Individual fixed effects	YES	YES	YES
Year fixed effects	YES	YES	YES
Observations	63,334	62,601	62,600
**Panel B: OLS estimates controlling for individual characteristics and weather factors**			
PM_2.5_	−0.0036 ***	0.0019 **	−0.0008
	(0.0014)	(0.0009)	(0.0008)
Individual characteristics	YES	YES	YES
Weather factors	YES	YES	YES
Individual fixed effects	YES	YES	YES
Year fixed effects	YES	YES	YES
Observations	56,725	56,640	56,639

**Notes:** This table reports OLS estimates from a regression in Equation (1). Panel A reports OLS estimates without controlling for individual characteristics and weather factors. Panel B reports OLS estimates controlling for individual characteristics and weather factors. Robust standard errors are in parentheses. ** and *** denote significance at 5% and 1% levels, respectively (the same below).

**Table 3 ijerph-20-02385-t003:** IV estimates of the influence of air pollution on physical, mental, and human health.

	(4)	(5)	(6)
**First-stage results**	PM_2.5_	PM_2.5_	PM_2.5_
Thermal inversion	0.0065 ***	0.0065 ***	0.0064 ***
	(0.0008)	(0.0008)	(0.0008)
**Second-stage results**	Physical health	Mental health	Human health
PM_2.5_	−0.1147 ***	−0.0450 **	−0.0797 ***
	(0.0294)	(0.0178)	(0.0175)
Individual characteristics	YES	YES	YES
Weather factors	YES	YES	YES
Individual fixed effects	YES	YES	YES
Year fixed effects	YES	YES	YES
KP rk LM statistic	59.109 ***	58.657 ***	58.585 ***
KP rk Wald F statistic	54.352	53.954	53.904
Observations	56,725	56,640	56,639

**Notes:** This table reports IV estimates from a regression in Equations (1) and (2). At first-stage estimates, the dependent variables are all PM_2.5_ concentration. At second-stage estimates, the dependent variables are physical, mental, and human health. ** and *** denote significance at 5% and 1% levels, respectively (the same below).

**Table 4 ijerph-20-02385-t004:** Replacing the core independent variable.

	(7)	(8)	(9)
**First-stage results**	SO_2_	SO_2_	SO_2_
Thermal inversion	0.0030 ***	0.0030 ***	0.0030 ***
	(0.0004)	(0.0004)	(0.0004)
**Second-stage results**	Physical health	Mental health	Human health
SO_2_	−0.2461 ***	−0.0963 **	−0.1703 ***
	(0.0629)	(0.0382)	(0.0376)
Individual characteristics	YES	YES	YES
Weather factors	YES	YES	YES
Individual fixed effects	YES	YES	YES
Year fixed effects	YES	YES	YES
KP rk LM statistic	68.003 ***	67.904 ***	67.888 ***
KP rk Wald F statistic	65.074	61.992	61.978
Observations	56,725	56,640	56,639

**Notes:** At first-stage estimates, the dependent variables are all SO_2_ concentration. At second-stage estimates, the dependent variables are physical, mental, and human health. ** and *** denote significance at 5% and 1% levels, respectively (the same below).

**Table 5 ijerph-20-02385-t005:** Replacing the IV.

	(10)	(11)	(12)
**First-stage results**	PM_2.5_	PM_2.5_	PM_2.5_
Thermal inversion (110 m−540 m)	0.0061 ***	0.0060 ***	0.0060 ***
	(0.0008)	(0.0008)	(0.0008)
**Second-stage results**	Physical health	Mental health	Human health
PM_2.5_	−0.1274 ***	−0.0635 **	−0.0961 ***
	(0.0388)	(0.0248)	(0.0237)
Individual characteristics	YES	YES	YES
Weather factors	YES	YES	YES
Individual fixed effects	YES	YES	YES
Year fixed effects	YES	YES	YES
KP rk LM statistic	59.859 ***	58.731 ***	58.748 ***
KP rk Wald F statistic	55.195	54.215	54.229
Observations	56,725	56,640	56,639

**Notes:** At first-stage estimates, the dependent variables are all PM_2.5_ concentration. At second-stage estimates, the dependent variables are physical, mental, and human health. ** and *** denote significance at 5% and 1% levels, respectively (the same below).

**Table 6 ijerph-20-02385-t006:** Readjusting the ratio of physical health to mental health.

	(13)	(14)	(15)	(16)
**First-stage results**	PM_2.5_	PM_2.5_	PM_2.5_	PM_2.5_
Thermal inversion	0.0065 ***	0.0065 ***	0.0065 ***	0.0065 ***
	(0.0008)	(0.0008)	(0.0008)	(0.0008)
**Second-stage results**	Human health (3:7)	Human health (4:6)	Human health (6:4)	Human health (7:3)
PM_2.5_	−0.0658 ***	−0.0728 ***	−0.0866 ***	−0.0935 ***
	(0.0156)	(0.0162)	(0.0193)	(0.0215)
Individual characteristics	YES	YES	YES	YES
Weather factors	YES	YES	YES	YES
Individual fixed effects	YES	YES	YES	YES
Year fixed effects	YES	YES	YES	YES
KP rk LM statistic	69.733 ***	69.733 ***	69.733 ***	69.733 ***
KP rk Wald F statistic	62.876	62.876	62.876	62.876
Observations	56,639	56,639	56,639	56,639

**Notes:** At first-stage estimates, the dependent variables are all PM_2.5_ concentration. At second-stage estimates, the dependent variables are human health with different ratios of physical health to mental health. *** denote significance at 1% levels (the same below).

**Table 7 ijerph-20-02385-t007:** Falsification test.

	Height	Height	Height	Height
	(17)	(18)	(19)	(20)
PM_2.5_	0.0383	−0.1365		
	(0.0805)	(0.1622)		
SO_2_			0.1220	−0.2772
			(0.2567)	(0.3289)
Individual characteristics	NO	YES	NO	YES
Weather factors	NO	YES	NO	YES
Individual fixed effects	YES	YES	YES	YES
Year fixed effects	YES	YES	YES	YES
Observations	49,408	44,654	49,408	44,654

**Notes:** In Columns (17) and (18), the independent variables are the PM_2.5_ concentration. In Columns (19) and (20), the independent variables are SO_2_ concentration.

**Table 8 ijerph-20-02385-t008:** Heterogeneity effects with age.

	Young (≤40)		Middle (40–60)		Old (>60)	
	Physical Health	Mental Health	Physical Health	Mental Health	Physical Health	Mental Health
	(21)	(22)	(23)	(24)	(25)	(26)
PM_2.5_	−0.0774 *	−0.0146	−0.1136 ***	−0.0289	−0.3685	−0.2596
	(0.0400)	(0.0253)	(0.0372)	(0.0225)	(0.3732)	(0.2555)
Individual characteristics	YES	YES	YES	YES	YES	YES
Weather factors	YES	YES	YES	YES	YES	YES
Individual fixed effects	YES	YES	YES	YES	YES	YES
Year fixed effects	YES	YES	YES	YES	YES	YES
Observations	12,112	12,106	28,443	28,406	13,602	13,559

**Notes:** In Columns (21), (23) and (25), the dependent variable is physical health. In Columns (22), (24) and (26), the dependent variable is mental health. The samples are grouped by age. * and *** denote significance at 10% and 1% levels, respectively (the same below).

**Table 9 ijerph-20-02385-t009:** Heterogeneity effects with education.

	Less Educated (≤9 Years)		More Educated (>9 Years)	
	Physical Health	Mental Health	Physical Health	Mental Health
	(27)	(28)	(29)	(30)
PM_2.5_	−0.1358 ***	−0.0513 **	−0.0517	−0.0247
	(0.0393)	(0.0232)	(0.0351)	(0.0234)
Individual characteristics	YES	YES	YES	YES
Weather factors	YES	YES	YES	YES
Individual fixed effects	YES	YES	YES	YES
Year fixed effects	YES	YES	YES	YES
Observations	45,689	45,606	10,697	10,696

**Notes:** In Columns (27) and (29), the dependent variable is physical health. In Columns (28) and (30), the dependent variable is mental health. The samples are grouped by education. ** and *** denote significance at 5% and 1% levels, respectively (the same below).

**Table 10 ijerph-20-02385-t010:** Heterogeneity effects with coast.

	Coastal Areas		Inland Areas	
	Physical Health	Mental Health	Physical Health	Mental Health
	(31)	(32)	(33)	(34)
PM_2.5_	−0.1615 **	−0.1058 **	−0.0880 ***	−0.0072
	(0.0642)	(0.0420)	(0.0312)	(0.0185)
Individual characteristics	YES	YES	YES	YES
Weather factors	YES	YES	YES	YES
Individual fixed effects	YES	YES	YES	YES
Year fixed effects	YES	YES	YES	YES
Empirical *p*-value	0.074 **			
Observations	25,349	25,310	31,373	31,327

**Notes:** In Columns (31) and (33), the dependent variable is physical health. In Columns (32) and (34), the dependent variable is mental health. The samples are grouped by coast. The empirical *p*-value is used to test the inland area versus coastal area difference in physical health through the 1000 times bootstrap approach. ** and *** denote significance at 5% and 1% levels, respectively (the same below).

**Table 11 ijerph-20-02385-t011:** Heterogeneity effects with household register.

	Urban Areas		Rural Areas	
	Physical Health	Mental Health	Physical Health	Mental Health
	(35)	(36)	(37)	(38)
PM_2.5_	−0.0322 *	−0.0015	−0.1874 ***	−0.0787 **
	(0.0177)	(0.0105)	(0.0601)	(0.0362)
Individual characteristics	YES	YES	YES	YES
Weather factors	YES	YES	YES	YES
Individual fixed effects	YES	YES	YES	YES
Year fixed effects	YES	YES	YES	YES
Empirical *p*-value	0.155 ***			
Observations	25,033	25,013	30,882	30,815

**Notes:** In Columns (35) and (37), the dependent variable is physical health. In Columns (36) and (38), the dependent variable is mental health. The samples are grouped by household register. The empirical *p*-value is used to test the urban area versus rural area difference in physical health through the 1000 times bootstrap approach. *, ** and *** denote significance at 10%, 5% and 1% levels, respectively (the same below).

**Table 12 ijerph-20-02385-t012:** LD estimates of influence of air pollution on physical, mental, and human health (six years).

	(39)	(40)	(41)
**First-stage results**	ΔPM_2.5_	ΔPM_2.5_	ΔPM_2.5_
Thermal inversion (current year)	0.0036 ***	0.0036 ***	0.0036 ***
	(0.0005)	(0.0005)	(0.0005)
**Second-stage results**	ΔPhysical health	ΔMental health	ΔHuman health
ΔPM_2.5_	−0.0470	−0.0739 ***	−0.0638 ***
	(0.0356)	(0.0248)	(0.0216)
ΔIndividual characteristics	YES	YES	YES
ΔWeather factors	YES	YES	YES
Year fixed effects	YES	YES	YES
KP rk LM statistic	51.930 ***	50.263 ***	50.240 ***
KP rk Wald F statistic	54.388	52.644	52.619
Observations	23,304	22,999	22,998

**Notes:** This table reports estimates from a regression following Equations (3) and (4) with a six-years difference (2010–2016 and 2012–2018). At first-stage estimates, the dependent variables are all ΔPM_2.5_ concentration. At second-stage estimates, the dependent variables are Δphysical health, Δmental health, and Δhuman health. Thermal inversion in the current year is used as an IV. *** denote significance at 1% levels (the same below).

**Table 13 ijerph-20-02385-t013:** LD estimates of influence of air pollution on physical, mental, and human health (eight years).

	(42)	(43)	(44)
**First-stage results**	ΔPM_2.5_	ΔPM_2.5_	ΔPM_2.5_
Thermal inversion (current year)	0.0080 ***	0.0080 ***	0.0080 ***
	(0.0005)	(0.0005)	(0.0005)
**Second-stage results**	ΔPhysical health	ΔMental health	ΔHuman health
ΔPM_2.5_	−0.0295	−0.0460 ***	−0.0373 ***
	(0.0205)	(0.0136)	(0.0117)
ΔIndividual characteristics	YES	YES	YES
ΔWeather factors	YES	YES	YES
Year fixed effects	YES	YES	YES
KP rk LM statistic	202.289 ***	204.183 ***	204.242 ***
KP rk Wald F statistic	223.997	226.325	226.398
Observations	12,195	12,139	12,138

**Notes:** This table reports estimates from a regression following Equations (3) and (4) with an eight-year difference (2010–2018). At first-stage estimates, the dependent variables are all ΔPM_2.5_ concentration. At second-stage estimates, the dependent variables are Δphysical health, Δmental health, and Δhuman health. Thermal inversion in the current year is used as an IV. *** denote significance at 1% levels.

## Data Availability

Data from this study are available upon request.

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
