# Peer review of "The Short- and Long-Run Impacts of Air Pollution on Human Health: New Evidence from China"

_ijerph, 2023, doi:10.3390/ijerph20032385_

Round 1

Reviewer 1 Report

The paper by Ren et al. discusses the effects of air pollution on human health. Overall, it is well written but has several grammatical errors/issues and confusing statements at certain locations. The paper uses a large dataset and tries to give a perspective on health based on correlation analysis. Though I appreciate the effort in the analysis, but I don’t agree with some of the conclusions drawn. I find the outcomes to be speculative. I would suggest a reanalysis of the data/parameters and also urge the authors to look into the papers/reports from other parts of the world to justify the findings.

Line 26-28: Please rephrase the sentence.

Line 31-36: Long, confusing sentence. Please rephrase it to make it clearer.

Line 37-47: Unclear and confusing paragraph. English used is very poor, which makes understanding difficult.

Line 50-52: Please check the format of PM2.5. The presented Pm values are from which location and period.

Line 64-67, 68-71: Please check the statement. I would advise not to use lengthy single statements.

Line 70-71: What is the meaning of physical, mental, and overall perspectives?

Line 107-111: How logical is the mental health measurement parameter considered here?

Line 114: Please check the representation of symbols here and in other places.

Line 124-133: Why are the findings discussed in the introduction section? Seems illogical.

Line 179-184: Please remove this paragraph.

Line 189-191: Please rephrase these statements to make them clearer.

Line 197-198: It should be ‘availability’ or ‘unavailability.’

Line 210-211: What does this line mean?

Line 257-259: How can one be so sure of the depression occurring and only due to air pollution?

Line 261-263:  How accurate is the rating-based average of mental and physical health?

Line 308-309: Please mention the reference to which these layer heights have been presented.

Line 346: Please avoid the term ‘confounder.’

Line 359-367: It is not clearly visible whether the estimated coefficients are positive, negative, or zero.

Line 408 and line 410: Please check the format of air pollution parameters here and in the entire manuscript.

Line 494: It would be better if 3 categories were considered: young (<40), middle-aged (40-60), and old (>60).

Line 540-550: Coastal areas are less polluted compared to inland regions due to sea winds and the removal of pollutants from the atmosphere. Also, living on the coast can improve mental health, as seen previously. I’m doubtful about the inferences drawn here.

Line 560-568: Rural areas usually represent background pollution due to fewer sources and population, and urban areas have high pollution due to development/economic activities. This can result in more severe health issues in the urban populations, as observed worldwide, but your finding shows the opposite. Please check this.

Line 570-571: Please rephrase this sentence.

Line 571-573: There is a good amount of literature showing the influence of air pollution on physical health in the long run. I would advise you to please see those. This statement is wrong.

Line 581: Please define the LD model.

Line 611-612, 625-626: This is a very doubtful outcome. I would suggest rechecking the calculation and parameters. Mental health can be degraded due to several other reasons, but I seriously doubt air pollution is one of the reasons.

Line 627-629: Several studies and medical reports have shown long-term physical health issues in highly polluted areas. I beg to differ on this outcome.

Line 636: But how can one be so sure that air pollution resulted in mental health issues which, according to this study, is only based on rating-based analysis? Whereas data on the degrading health effects, in the long run, are more reliable from various renowned agencies.

Line 637-643: In my opinion, these inferences are speculative and not backed by proper data/analysis.

Section 6: It is unclear how the health cost is computed?

Line 671-672: Confusing statement.

Reviewer 3 Report

Dear Authors ,

I have now completed my assessment of the manuscript titled “Influence of air pollution on human health: New evidence from  China”.

However, following are some of my comments which I feel the authors must address before it can be accepted for publication in the Sustainability journal:

1. The title should attract the audience. 2. The abstract should state in some detailed form for the purpose of the research, the principal results, and major conclusions. 3. The current study lacks motivation. Authors should expand the main goals and objectives of this work to improve on the motivation of the current study. The novelty of this work is not clear. 4. I would like to suggest that authors should update the literature part. Specifically, the latest research trends, and in order to highlight the academic frontier of the research, the references of the recent year need to be referenced. In this case, the following paper may be helpful:

10.1007/s11356-022-22011-1

5. I'll advise authors to separate the "Literature review and Hypothesis development" section into its own section. 6. More importantly, per my critical observation, the authors in this manuscript reported results from the analysis without discussing them. I, therefore, suggest that authors should instead separate the presentation of results from the discussions. Both should be in different sub-sections under the title "Results and discussions. Moreover, the authors are supposed to give the impact mechanism behind the results obtained (that is, detailed economic meaning of their results and implications or, in other words, what might have brought about the results they have obtained). It must be done systematically by comparing them to further studies in the related fields. 7. The policy recommendation is too slim. Authors should add more to this section, especially in the aspect of policy framing and implementation. 8. Method seems fine to me. 9. There are several grammatical issues throughout the paper. Please ask a professional copy-editing company for the English editing. 10. Modify the paper based on the journal’s guidelines.

Round 2

Reviewer 2 Report

I congratulate the authors for all the changes and improvements they made to the manuscript. I really think the suggestions of the reviewers make it a lot more complete and clear.